# Biomechanical risk factors for knee osteoarthritis and lower back pain in lower limb amputees: protocol for a systematic review

Logan Wade , M Polly McGuigan, Carly McKay, James Bilzon, Elena Seminati

Department for Health, University of Bath, Bath, UK

**Correspondence to**
Dr Logan Wade;
lw2175@bath.ac.uk

## ABSTRACT

**Introduction** There is a limited research exploring biomechanical risk factors for the development of knee osteoarthritis (KOA) and lower back pain (LBP) between lower limb amputee subgroups, (eg, transtibial amputees (TTA) vs transfemoral amputees (TFA), or TTA dysvascular vs TTA traumatic). Previous reviews have focused primarily on studies where symptoms of KOA or LBP are present, however, due to limited study numbers, this hinders their scope and ability to compare between amputee subgroups. Therefore, the aim of this systematic review is to descriptively compare biomechanical risk factors for developing KOA and LBP between lower limb amputee subgroups, irrespective of whether KOA or LBP was present.

**Methods and analysis** This review is currently in progress and screening results are presented alongside the protocol to highlight challenges encountered during data extraction. Five electronic databases were searched (Medline—Web of Science, PubMed, CINAHL, Embase and Scopus). Eligible studies were observational or interventional, reporting biomechanical gait outcomes for individual legs in adult lower limb amputees during flat walking, incline/decline walking or stair ascent/descent. Two reviewers screened for eligibility and level of agreement was assessed using Cohen's Kappa. Data extraction is ongoing. Risk of bias will be assessed using a modified Downs and Black method, and outcome measures will be descriptively synthesised.

**Ethics and dissemination** There are no ethical considerations for this systematic review. Due to its scope, results are expected to be published in three separate manuscripts: (1) biomechanical risk factors of KOA between TTA and TFA, relative to non-amputees, (2) biomechanical risk factors of LBP between TTA and TFA, relative to non-amputees and (3) biomechanical risk factors of KOA and LBP between TTA with traumatic or dysvascular causes, relative to non-amputees.

**PROSPERO registration number** CRD42020158247.

## STRENGTHS AND LIMITATIONS OF THIS STUDY

⇒ This systematic review protocol follows the Preferred Reporting Items for Systematic Review and Meta-Analysis Protocols guidelines.
⇒ Biomechanical gait will be compared between amputee subgroups (transtibial vs transfemoral amputees, and transtibial dysvascular vs transtibial traumatic amputees).
⇒ Studies must include at least one temporospatial, joint kinematic or joint kinetic outcome measure for individual legs.
⇒ Only amputee studies that included non-amputee controls will be included in the systematic review.

## INTRODUCTION

Lower limb amputations of the hip, knee and ankle considerably alter walking gait and function, with over 42 000 major lower limb amputations performed over a 10-year period (2003–2013) in the UK.[1] In 2005, major lower limb amputations in the USA and UK accounted for over 90% of all major limb amputations[2 3] and compared with healthy populations, lower-limb amputees have significantly higher rates of secondary disorders such as knee osteoarthritis (KOA)[4 5] and lower back pain (LBP).[6–11] While there are many biopsychosocial factors that may contribute to the higher rates of secondary disorders (eg, mental health, diet, access to facilities or social organisations), the biomechanical factors which result in altered gait of amputee populations will potentially also play a major role.[12] Stable lower limb amputee gait often requires the intact leg to support greater load, which introduces gait asymmetries that over the lifetime, may result in overuse and greater wear of joints and muscles compared with non-amputees. Furthermore, differences between amputation levels (below ankle, below knee and above knee) and amputation causes (traumatic, vascular disease, cancer, congenital) may produce different functional impairments, which could increase the risk of developing KOA and LBP in these different amputee populations.

Considering the prevalence of lower limb amputations, transfemoral (above the level of the knee) amputees (TFA) and through

knee (at the level of the knee joint) amputees account for 17%–23% of all amputations.[13 14] Transtibial (below the level of the knee) amputees (TTA) and through ankle (at the level of the ankle joint) amputees account for 12%–32%, while partial foot amputees account for 15%–26% of all amputations.[13 14] Minor amputations of the foot make up the remaining percentages, however, these generally do not substantially alter gait and are therefore not a focus of this review. As amputation level moves up the leg, functional mobility and quality of life is reduced,[15] requiring greater altered gait mechanics to accommodate the limited power output and instability of the prosthetic limb during stance.[12] Thus, above knee amputees are at an increased risk of developing knee pain[4] and KOA in the intact limb compared with below knee amputees, with OA of the intact knee occurring in roughly 60% of TFAs and 40% of TTAs, compared with just 20% of non-amputees.[16] Similarly, prevalence of LBP is found in roughly 50%–76% of lower limb amputees, compared with 35% of non-amputees.[10–12] Evidence suggests that there may not be a difference in prevalence or intensity of LBP between TTA and TFA,[17] although a previous systematic review of LBP in lower limb amputees was unable to draw comparisons between TTA and TFA due to limited studies in TTA.[18] Thus, there is a need to explore biomechanical gait differences between TTAs and TFAs, to understand how biomechanical risk factors associated with the development of and potential predisposition to KOA and LBP differ between groups.

While amputation level plays a crucial role in altered gait mechanics, cause of amputation likely also contributes significantly to the development of secondary musculoskeletal symptoms. The two primary causes of amputation are vascular diseases and traumatic accidents, with cancer and congenital causes only making up 1%–3% of all amputations.[3 14] Prevalence of amputation cause varies worldwide, with traumatic amputations making up 6%–45% of all amputations[3 14] and patients primarily characterised as being young and fit.[3] Alternatively, dysvascular amputations have increased significantly in recent decades due to the increasing prevalence of diabetes and dysvascular disease, making up 65%–91% of all amputations.[3 14] This population is generally older than other amputee cause types[3] and commonly have a higher body mass index,[19] which additionally puts individuals at a greater risk of KOA.[20] Dysvascular amputees also have poorer uptake of prosthetic devices, which further increases their risk of sedentary lifestyle and weight gain after amputation.[21] Counterintuitively, some research suggests that this lower activity status and prosthetic use may result in TFAs having a reduced risk of developing LBP compared with traumatic amputees.[16 18] Unfortunately, despite a much higher prevalence of dysvascular amputations, gait biomechanics research within this population is relatively limited, especially compared with the high proportion of research surrounding traumatic amputations.[4 11 18 22–25] We therefore need to determine whether current research, investigating the development

of secondary disorders primarily in traumatic amputees, is generalisable to dysvascular amputees, and if there are any additional biomechanical factors specific to dysvascular amputees that would increase or decrease their likelihood of developing KOA and LBP.

Additional subgroups include bi-lateral amputees, osseo-integrated amputees and adult amputees who had an amputation as children or were born without a limb (ie, congenital amputees). Bilateral amputees have a high variation between individuals, often presenting with multiple amputation levels (eg, one leg with a TTA and the other with a TFA), which can dramatically alter gait and may influence development of secondary disorders. Osseo-integrated amputees generally do not suffer from skin problems, ill-fitting prosthesis issues or bone degeneration issues of their socket wearing counterparts. Thus, this population may have greater prosthetic use and increased risk of KOA and LBP, although they also have alternate complications such a recurring infections and risk of bone fractures.[26 27] Finally, adult amputees who experienced amputations during childhood, or were congenital amputees, have spent the most time with their amputation. This group may have altered gait patterns as a function of growing with their prosthesis, which may place them at an increased risk of developing secondary symptoms much earlier in life. Across all amputee subgroups, the primary barrier to understanding altered biomechanical gait is in recruiting a sufficient sample from each population, especially in these latter specialised subgroups. Furthermore, longitudinal cohort studies, following patients throughout their life are very rare, with most studies being performed cross-sectionally. Therefore, a large-scale systematic review that examines biomechanical gait between amputee subgroups is presently the best available option for exploring which biomechanical gait factors may contribute to development of KOA or LBP between lower limb amputee populations.

Several reviews have examined amputee biomechanical gait with a focus on KOA and LBP. However, the majority of these reviews have not been performed using systematic methods,[11 22 23 28–30] and generally have not described differences between amputee subgroups, often only including a single subgroup (eg, only traumatic or TTA). Moreover, those few systematic reviews on gait and secondary disorders in amputees have generally only been performed on a single amputee subgroup, using studies where symptoms of KOA or LBP are present, which severely limits their scope (11–17 studies per review) and ability to compare between amputee groups.[16 18 31 32] Due to such small study numbers included within these systematic reviews, knowledge of the biomechanical gait characteristics associated with KOA and LBP and their prevalence between amputee subgroups is considerably limited. Sagawa et al[33] has performed a large-scale systematic review (89 studies) of altered biomechanical gait factors across all lower limb amputees, aiming to broadly characterise biomechanics and

physiological parameters during gait. They identified that TTA knee flexion during heel strike is limited to 9°–12°, while TFA knee flexion was zero or negative (extension). Additionally, TFAs had two times the pelvic range of motion compared with healthy individuals which may contribute to the development of LBP. Unfortunately, their review was very broad, was not targeted at gait characteristics of KOA and LBP and generally did not make any comparisons or conclusions between subgroups (eg, amputation level or amputation cause). To fill this gap in the literature, a large-scale systematic review targeted at identifying how biomechanical risk factors of KOA and LBP differ between amputee subgroups is needed. Understanding what biomechanical factors influence gait will help facilitate specific and personalised rehabilitation programmes and prosthetic designs.

## Objectives

While previous systematic reviews have been limited by only including studies with amputees who are diagnosed with KOA and LBP, there is a substantial amount of experimental literature that has examined lower limb amputee gait and posture where no KOA or LBP has been recorded. Because of the high prevalence of KOA and LBP, it is likely that biomechanical abnormalities leading to these secondary disorders will be present across the majority of amputees. Therefore, the aim of this systematic review is to descriptively compare biomechanical risk factors for developing KOA and LBP between amputee subgroups, irrespective of whether KOA or LBP was present. Amputee subgroups will be categorised by level of amputation (below ankle, below knee and above knee), cause of amputation (vascular disease, traumatic injury, cancer, congenital) and special subgroups (bilateral amputees, osseo-integrated amputees and adult amputees who had an amputation or congenital missing limb as children). Individual subgroups will only be included for analysis if sufficient data is available to support comparisons (see the Data extraction section).

## METHODS

This systematic review is currently in progress with the first search completed on 3 July 2017 and a projected end date of 1 December 2023. Screening results are presented within this paper to highlight challenges encountered during data extraction. This approach was chosen to ensure the transparency of our methods and increase the replicability of the review.

## Eligibility criteria

In accordance with Preferred Reporting Items for Systematic Review and Meta-Analysis Protocols (PRISMA-P) guidelines,[34] this protocol was submitted and approved by the International Prospective Register of Systematic reviews on 3 February 2020 and was last updated 21 January 2022. This protocol has adhered to the PRISMA-P guide and checklist for publishing systematic review protocols.[34]

## Study characteristics

Studies included in this review had to be observational studies such as cross-sectional/cohort studies and longitudinal studies. Intervention and randomised control trial studies were included in this review but only the control amputee group or baseline measures were extracted (observational data). Review papers, case studies, conference proceedings and animal studies were excluded. Studies that included quantitative biomechanical measures of lower limb amputees were included if results were reported for individual legs (intact leg and prosthetic leg presented separately). To ensure application of valid and thorough biomechanical technique and analysis, data had to include at least one temporospatial, joint kinematic or joint kinetic outcome measure for individual legs (see online supplemental appendix 1 for a full list of extracted outcome measures). Outcome variables were determined from previous reviews that outlined biomechanical differences between: amputees and non-amputee populations[12 17 22 23 28 33 35]; healthy non-amputee populations and KOA and LBP non-amputee populations[36–38]; and healthy amputees and amputees with KOA and LBP.[12 16 18 31 32] While ground reaction force (GRF) outcome measures for individual strides were extracted, studies that only reported GRF measures were not included in this review, as GRF is a measure of full body force and is not specific to the knee joint or lower back region. Observational studies had to be performed during walking on flat, incline or stair surfaces, at either preferred or controlled walking speeds. Studies that only investigated running-specific prostheses or running gaits were not included. Studies that examined powered ankles were included in this review, but only if an unpowered condition was performed. All microprocessor-controlled ankles and knees (devices that do not add energy to the system) were included in this review.

## Participants

Lower limb amputees were included in this review, but only if results were separated by different amputation levels (eg, studies that combined results of TTA and TFA were not included). Due to the differences between child and adult gait, and the focus on development of secondary disorders which primarily occurs in adults, studies performed only on children (younger than 18 years) were not included.

## Patient and public involvement
None.

## Information sources

Literature searches were performed across five databases: Medline—Web of Science, PubMed, CINAHL, Embase and Scopus. Manual searches were conducted using the reference lists within previous reviews and reference lists

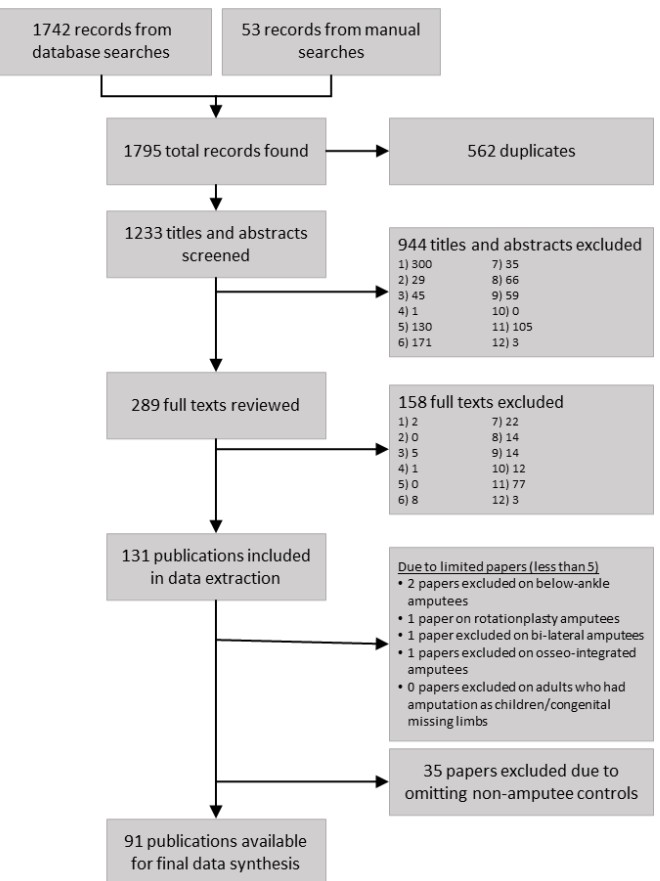

**Figure 1** Flow chart of paper selection. Exclusion reasons are: (1) no amputees, (2) upper limb amputation, (3) no adult human participants, (4) language not English, (5) review, (6) no quantitative data, (7) paper not found/duplicate, (8) no clinical outcomes, (9) single case study), (10) no results for separate amputee groups, (11) no biomechanical parameters, (12) powered prosthesis only.

within papers obtained from database searches, to ensure all relevant literature was identified (figure 1).

### Search strategy

Studies were only examined if they were published in English. Only peer-reviewed studies were included. No publication date limit was imposed on the search criteria. Search terms included a combination of amputation terms AND gait/biomechanics terms AND secondary disorders. While inclusion for this systematic review did not require the presence of secondary disorders, this term helped to refine the search and identify papers with outcome measures of relevance to the development of secondary disorders in amputee populations. An example search strategy is presented below and a table of the full search strategy, formatted for each database, can be found in online supplemental appendix 2.

1. Amputee: "transtibial amput*" OR "transfemoral amput*" OR amput* OR "Lower limb amput*" OR "Lower extremity amput*" OR "Leg prosthesis".
2. Activity: walking OR running OR gait OR locomotion OR biomechanics OR kinematics OR kinetics OR

"biomechanical parameter*" OR *symmetr* OR forc* OR angle* OR moment* OR power EMG OR electro-myogra*).
3. Secondary disorder: Osteoporosis OR Osteopenia OR "Back Pain" OR Backache OR Osteoarthritis OR "musculoskeletal diseas*" OR "musculoskeletal condition*" OR "secondary diseas*".

### Data management and selection process

Records retained for abstract and full paper screening were compiled using an excel spreadsheet designed for systematic reviews.[39] Two reviewers individually applied the eligibility criteria to all records based on the inclusion/exclusion criteria outlined in figure 1. Where conflicts arose, reviewers met to discuss and if agreement still could not be made, a third reviewer was consulted to make the final decision. Review stages progressed from title and abstract review to full paper review (figure 1). For the title and abstract stage, there were four reviewers, with one person reviewing all papers and the remaining three people each reviewing a third of the papers. For the full paper stage, there were three reviewers, with one person reviewing all papers and the remaining two people reviewing half of the papers each. Level of agreement was assessed using Cohens Kappa.[40] Agreement for the title and abstract review stage was 0.76, while agreement for the full paper review stage was 0.64, where agreement between 0.61 and 0.80 represents substantial agreement between reviewers. A minimum of five studies that evaluated a specific subgroup were required to be included for evaluation of said subgroup within this systematic review. Due to a limited number of papers included after full-text review, studies that examined below ankle amputation (two papers), rotationplasty amputation (one paper), bi-lateral amputation (one paper), osseo-intregration (one paper) and adult amputees who had an amputation or congenital missing limb as children (0 papers) were ultimately excluded.

### Current stage

This systematic review is currently at the stage of performing data extraction.

### Data collection process

Data are currently being extracted from studies using a standardised excel spreadsheet. All data are being extracted by a single reviewer to ensure consistency, though a random sample of 20% of the data are also being extracted by a second reviewer to assess the risk of bias in the extraction process. Where necessary, extraction from figures is being been performed using the desktop version of WebPlotDigitizer,[41] which is a data extraction tool for plots, images and maps.

### Data items

Data items being extracted include manuscript title, authors, journal, year, country where data was collected, study type, amputee population, number of participants, amputation level, age, biological sex, body mass, height,

time since amputation, cause of amputation, type of prosthetic, years of prosthetic use, secondary symptoms, tasks performed in the study and outcome variables (temporospatial, joint kinematics and joint kinetics). For a detailed list of all biomechanical outcome variables, see online supplemental appendix 1. Mean/median values, along with SD/ranges are being extracted. For intervention studies, only the baseline measure will be extracted, thus all data included within this review will be observational and cross-sectional in nature.

During data extraction, it has become evident that some outcome measures may appear very high or very low for both amputee and non-amputee groups within the same study. For example, Hendershot and Wolf[42] examined trunk angle during walking gait using inverse dynamics, identifying that maximum extension for TTA was 4.89°, TFA was 0.48° and non-amputee controls were 2.75°. Morgenroth *et al*[43] also examined trunk angle during walking, however, their analysis was based on angle changes of a rigid cluster placed on the eighth thoracic vertebra (T8), with angles relative to the global coordinate system. Thus, they reported that maximum trunk extension of TFA was 26.9° while non-amputee controls were 20.5°. If absolute values were compared, the large maximum angles obtained for TTA's by Morgenroth *et al*[43] would drastically alter the differences observed between TTA and TFA across all studies. Therefore, studies which did not examine paired amputee groups (TTA vs TFA or vascular vs traumatic) have the potential to drastically alter the results, due to methodological differences in how data were collected. However, if studies recruited both amputees and non-amputees, relative differences compared with non-amputees within the same study could be calculated. Using the example above for Hendershot and Wolf,[42] relative maximum trunk angle in TTA was 2.1° larger than non-amputee controls and TFA was 2.3° smaller than non-amputee controls, while Morgenroth *et al*[43] observed TFA was 6.4° larger than non-amputee controls. Unfortunately, if studies only recruited amputees and did not recruit non-amputee controls, calculation of relative differences between amputees and non-amputees cannot be calculated. The diverse range of methodologies included within this review was unexpected and only determinable due to this systematic review collating the largest number of biomechanical gait studies performed on amputees to date. Therefore, to ensure rigorous and objective comparison of outcomes between amputee subgroups, we have removed 27 studies from screening that did not recruit non-amputee controls (figure 1), excepting those studies that compared directly between TTA and TFA, or between dysvascular TTA and traumatic TTA. Challenges we are facing during data extraction highlight the key role non-amputee controls play during examination of amputee gait, and therefore, studies wishing to compare their results to prior research should recruit non-amputee participants to facilitate such comparisons.

## Future stages

All remaining stages of the protocol encompass the future work yet to be started, with major stages including risk of bias assessment and data synthesis.

## Outcomes and prioritisation

The primary outcomes will be the biomechanical variables listed in online supplemental appendix 1. Reporting of outcome measures will be grouped based on whether previous evidence suggests they may contribute to KOA or LBP. Kinetic measures not already normalised to body mass will be converted to enable comparison between studies. Mean/median outcome measures, relative to controls within the same study, will be compared between amputee groups (TTA vs TFA and traumatic vs dysvascular). To directly compare outcome measures between studies for KOA or LBP, measures will be grouped depending on the type of movement: preferred speed flat walking, controlled speed flat walking, preferred speed incline/decline walking, controlled speed incline/decline walking, preferred speed stair climbing or controlled speed stair climbing. These movements were selected as they are commonly performed in daily living and present different challenges for amputees. Thus, to examine differences between amputation level, outcome measures related to KOA or LBP will be descriptively compared during each movement, between TTA and TFA, relative to non-amputees. To examine differences between amputation cause, outcome measures related to KOA or LBP will be descriptively compared during each movement, between transtibial traumatic and transtibial dysvascular amputees, relative to non-amputees.

## Risk of bias in individual studies

Risk of bias will be assessed using the modified Downs and Black method.[44 45] In this modified version, question 25 which addresses sample size, will be modified to a yes/no question and studies that performed a sample size calculation/power calculation will be awarded one point, while studies without will be awarded zero.[44] Randomised controlled trials will be assessed separately to reduce the impact of increased weighting placed on these studies by the Downs and Black method. Randomised controlled trials will only have baseline outcome measures extracted, so while risk of bias will be analysed separately for observational and intervention studies, outcome measures and presentation of the data will be performed identically across all studies. Two reviewers will both assess each study using the Downs and Black criteria. Where there are conflicts, reviewers will meet to discuss and if they cannot agree, a third reviewer will be consulted to make a final decision.

## Data synthesis and dissemination

The primary goal of this systematic review is to descriptively compare biomechanical risk factors for developing KOA and LBP between amputee subgroups, irrespective of whether KOA or LBP was present. Due to such a

large combination of outcome measures (online supplemental appendix 1), subgroups and gait types, meta-analyses will not be performed. Instead, quantitative results will be synthesised and descriptively compared using biomechanical mean/median values of amputee subgroups relative to non-amputees. Due to the scope of this review, results are expected to be published in three separate manuscripts: (1) biomechanical risk factors of KOA between TTA and TFA, relative to non-amputees, (2) biomechanical risk factors of LBP between TTA and TFA, relative to non-amputees and (3) biomechanical risk factors of KOA and LBP between TTA with traumatic or dysvascular causes, relative to non-amputees. KOA and LBP will be grouped in the third results paper, as there are far fewer studies that have solely recruited dysvascular amputees. The quality of evidence for all outcomes will be judged using the Grading of Recommendations Assessment, Development and Evaluation working group methodology. Systematic review analysis and reporting will be performed using the Preferred Reporting Items for Systematic Reviews and Meta-Analyses guidelines.[46]

## Meta-analysis and meta-bias

Due to the high number of movements (eg, walking, incline walking, decline walking), subgroups (eg, TFA, TTA, dysvascular and traumatic amputation) and outcome variables (temporospatial, kinematic and kinetic measures), which significantly reduces the number of studies that are able to be statistically compared for each outcome measure, a meta-analysis will not be performed. Therefore, examination of meta-bias within this review is not possible.

**Contributors** LW is the guarantor. LW, MPM, CM, JB and ES contributed to conception and design of the study. ES and CM developed the search strategy. LW, MPM, CM and ES and performed study selection. LW drafted the manuscript. LW, MPM, CM, JB and ES read, provided feedback and approved the final manuscript.

**Funding** This work was supported by the Engineering and Physical Sciences Research Council, through the RCUK Centre for the Analysis of Motion, Entertainment Research and Applications (CAMERA), Bath, UK, grant number (EP/M023281/1, EP/T014865/1). This work was also supported by the Versus Arthritis Centre for Sport Exercise, Exercise and Osteoarthritis Research (Ref 21595).

**Competing interests** None declared.

**Patient and public involvement** Patients and/or the public were not involved in the design, or conduct, or reporting, or dissemination plans of this research.

**Patient consent for publication** Not applicable.

**Provenance and peer review** Not commissioned; externally peer reviewed.

**ORCID iD**
Logan Wade http://orcid.org/0000-0002-9973-9934

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
