## [Reviewer comments · BMJ Open]

ARTICLE DETAILS

TITLE (PROVISIONAL)	Biomechanical risk factors for knee osteoarthritis and lower back pain in lower limb amputees: protocol for a systematic review
AUTHORS	Wade, Logan; McGuigan, M. Polly; McKay, Carly; Bilzon, James; Seminati, Elena

VERSION 1 – REVIEW

REVIEWER	Hendrick, Paul University of Nottingham
REVIEW RETURNED	18-Oct-2022

GENERAL COMMENTS	This manuscript presents a review protocol for a systematic review of biomechanical risk factors for knee OA and LBP in subgroups of lower limb amputees The rationale, background and methods are clearly presented and a clear rationale for the review is discussed My only comments would be that 1. It is unclear why differences in biomechanical variables between amputees and non-amputees can be classified as risk factors for the development of pain. The authors recognise the lack of prospective research in this area and so these factors would really be 'potential' or 'putative' in relation to factors for either knee OA and or LBP2. The authors do not really discuss why only biomechanical factors are focused on within a biopsychosocial model and perhaps this could be highlighted
---

VERSION 1 – AUTHOR RESPONSE

My only comments would be that

It is unclear why differences in biomechanical variables between amputees and non-amputees can be classified as risk factors for the development of pain. The authors recognise the lack of prospective research in this area and so these factors would really be 'potential' or 'putative' in relation to factors for either knee OA and or LBP.

REPLY: Thank you for your comment. We have softened the first paragraph to highlight that these biomechanical variables have the 'potential' to lead to development of pain. Instead of being likely to lead to pain, as was stated in the original manuscript. Additionally, we have expanded parts of this paragraph to better clarify how differences in biomechanical factors between amputees and non-amputees could be classified as increased risk factors of developing secondary disease.

Line 69:

While there are many biopsychosocial factors that may contribute to the higher rates of secondary disorders, (e.g. mental health, access to facilities or social organisations, etc), the biomechanical factors which result in altered gait of amputee populations will potentially also play a major role ¹². Stable lower limb amputee gait often requires the intact leg to support greater load, which introduces gait asymmetries that over the lifetime, may result in overuse and faster wearing joints compared to non-amputees.

The authors do not really discuss why only biomechanical factors are focused on within a biopsychosocial model and perhaps this could be highlighted

REPLY: Thank you for your comment, we do agree that acknowledging additional factors that are likely contributing to development of secondary disorders is important. As such we have added a sentence to the introduction to highlight that biomechanical factors alone are likely not the only cause of developing secondary disorders such as knee osteoarthritis and lower back pain.

Line 69:

While there are many biopsychosocial factors that may contribute to the higher rates of secondary disorders, (e.g. mental health, access to facilities or social organisations, etc), the biomechanical factors which result in altered gait of amputee populations will potentially also play a major role

VERSION 2 – REVIEW

REVIEWER	Hendrick, Paul University of Nottingham
REVIEW RETURNED	03-Nov-2022

GENERAL COMMENTS	I would like to thanks the authors for addressing the reviewer comments .
---